# Cerebellar Cavernoma Resection: Case Report with Long-Term Follow-Up

**DOI:** 10.3390/jcm13247525

**Published:** 2024-12-11

**Authors:** Corneliu Toader, Matei Serban, Razvan-Adrian Covache-Busuioc, Mugurel Petrinel Radoi, Ghaith Saleh Radi Aljboor, Horia Petre Costin, Antonio Daniel Corlatescu, Luca-Andrei Glavan, Radu Mircea Gorgan

**Affiliations:** 1Department of Neurosurgery, “Carol Davila” University of Medicine and Pharmacy, 050474 Bucharest, Romania; corneliu.toader@umfcd.ro (C.T.); razvan-adrian.covache-busuioc0720@stud.umfcd.ro (R.-A.C.-B.); petrinel.radoi@umfcd.ro (M.P.R.); ghaith-saleh-radi.aljboor@rez.umfcd.ro (G.S.R.A.); horia-petre.costin0720@stud.umfcd.ro (H.P.C.); antonio.corlatescu0920@stud.umfcd.ro (A.D.C.); luca-andrei.glavan0720@stud.umfcd.ro (L.-A.G.); radu.gorgan@umfcd.ro (R.M.G.); 2Department of Vascular Neurosurgery, National Institute of Neurology and Neurovascular Diseases, 077160 Bucharest, Romania; 3Department of Neurosurgery, Clinical Emergency Hospital “Bagdasar-Arseni”, 041915 Bucharest, Romania

**Keywords:** cerebellar cavernoma, cerebral cavernous malformation, hemorrhagic cavernoma, microsurgical resection, postoperative recovery, long-term follow-up

## Abstract

**Background**: Cerebral cavernous malformations (CCMs), particularly when located in the cerebellum, pose unique clinical challenges due to the risk of hemorrhage and proximity to critical neurovascular structures. Surgical resection is often necessary to prevent further neurological deterioration. This case report describes the management of a symptomatic cerebellar cavernoma, emphasizing the use of microsurgical techniques and long-term follow-up. The objective of this study is to illustrate the surgical approach and outcomes of a patient with a hemorrhagic cerebellar cavernoma. **Methods**: A 63-year-old female presented with vertigo, and neuroimaging revealed a hemorrhagic cavernous malformation located in the right cerebellar hemisphere. Follow-up was conducted at two months and one year postoperatively, with serial imaging to assess lesion recurrence and neurological recovery. **Results**: Complete resection of the cavernoma was achieved without postoperative neurological deficits. Imaging at two months and one year post-surgery showed no signs of recurrence or new lesion formation. The patient remained asymptomatic, with no cranial nerve deficits or other long-term complications. **Conclusions**: This case demonstrates the effectiveness of microsurgical resection in treating symptomatic cerebellar cavernomas. The use of advanced intraoperative tools, such as neuronavigation and IONM, contributed to the successful outcome and prevention of postoperative complications. Long-term follow-up remains crucial to monitor for recurrence or the development of de novo lesions.

## 1. Introduction

Cerebral cavernous malformations (CCMs) are vascular lesions composed of clusters of dilated capillaries with slow blood flow, often resulting in recurrent microhemorrhages. These malformations are well recognized for their potential to cause significant neurological deficits when located in critical regions of the brain. Although CCMs account for approximately 10–15% of all intracranial vascular malformations, their presence in the posterior fossa—particularly in the cerebellum—accounts for roughly 15–20% of all intracranial cases, making them relatively rare in this location [1].

Recent advancements in neuroimaging, such as Arterial Spin Labeling (ASL), have enhanced the diagnosis and follow-up of CCMs. ASL provides quantitative, non-invasive cerebral perfusion data, complementing T2-weighted and susceptibility-weighted imaging (SWI) [2]. Particularly in posterior fossa lesions, ASL aids in detecting changes in perfusion patterns that may indicate lesion activity, microhemorrhages, or recurrence, offering a valuable tool for long-term monitoring and management [3].

Patients with cerebellar cavernomas typically present with vertigo, ataxia, headaches, and cranial nerve deficits, depending on the lesion’s location and proximity to the brainstem. Notably, cerebellar cavernomas carry an increased risk for hemorrhage compared to those in supratentorial regions, with studies indicating annual hemorrhage rates between 4.1% and 9.8%, depending on clinical factors such as previous hemorrhage and lesion size. As a result, early diagnosis and management are critical in preventing significant neurological decline and recurrent bleeding, which can lead to permanent deficits [4]. Advances in neuroimaging, particularly magnetic resonance imaging (MRI), have enhanced the detection and characterization of cavernomas. MRI, particularly T2-weighted and SWI, is the gold standard for identifying these lesions and their associated hemosiderin deposits, providing detailed information on the lesion’s size, structure, and prior hemorrhage [5]. Despite these advances, management remains challenging, especially for symptomatic posterior fossa cavernomas, where the anatomical complexity of the region can pose significant risks during surgical intervention [6].

Surgical resection is generally recommended for symptomatic patients or those with evidence of hemorrhage. However, the intricate neurovascular structures in the posterior fossa make surgery technically demanding. Furthermore, the introduction of novel hemostatic agents and precise intraoperative techniques has improved patient outcomes, with several studies demonstrating complete resection rates above 90%, with minimal postoperative morbidity [7].

To further refine management strategies, the posterior cranial fossa cavernoma grading system (PCFCGS) has been developed to classify these lesions based on key factors influencing surgical risk and outcomes. This system considers variables such as lesion size, proximity to critical neurovascular structures like the brainstem, and a history of hemorrhage [8]. Higher-grade lesions are associated with increased surgical complexity and a greater likelihood of postoperative complications, necessitating careful preoperative evaluation and planning [9]. By stratifying risk, the PCFCGS provides a standardized framework that assists clinicians in making informed decisions, particularly in cases of symptomatic posterior fossa cavernomas, where the stakes of intervention are exceptionally high [10].

Long-term follow-up is essential for patients undergoing resection of cavernomas, particularly those located in the posterior fossa. Studies have shown that while recurrence rates after complete resection are low, patients may develop de novo lesions or experience late-onset complications [11]. Thus, regular surveillance using MRI is recommended, with some authors advocating annual imaging for the first few years postoperatively, followed by less frequent long-term surveillance [12,13].

In this case report, we present a rare cerebellar cavernoma in a 63-year-old female who experienced debilitating vertiginous symptoms. Despite the lesion’s challenging posterior fossa location—adjacent to critical neurovascular structures—a meticulous microsurgical approach achieved complete en bloc resection. The patient’s postoperative course was remarkable, with full neurological recovery and no recurrence on follow-up imaging. The novelty of this case lies in the integration of advanced surgical techniques to manage a deeply seated hemorrhagic cavernoma in one of the most anatomically complex regions of the brain. This report further highlights the importance of precision-driven interventions and contributes to the limited literature on long-term outcomes following posterior fossa cavernoma resections, offering new perspectives on optimizing patient care in similar high-risk scenarios.

## 2. Case Presentation

A 63-year-old female patient with a known history of hypertension presented to our clinic with a chief complaint of vertiginous syndrome. The symptoms gradually progressed over the course of several weeks, with the patient experiencing persistent dizziness and frequent episodes of imbalance. She denied any associated symptoms such as nausea, vomiting, hearing loss, or tinnitus. The patient’s medical history was notable only for well-controlled hypertension. She had no history of prior neurological disorders, head trauma, or infections. Her family history was unremarkable, and she had no known allergies. The patient was not taking any medications other than antihypertensive therapy.

On physical examination, the patient was alert and oriented to time, place, and person, with clear and coherent speech. Her vital signs were stable, and her blood pressure was measured at 145/85 mmHg. The neurological examination was largely normal, except for findings relevant to her vertiginous symptoms. Examination of cranial nerve VIII (vestibulocochlear) revealed subjective dizziness and imbalance. The imbalance observed in this case is definitively attributed to the involvement of the cerebellar peduncle rather than the vestibulocochlear nerve. The lesion was located in the right cerebellar hemisphere, an area directly connected to the cerebellar peduncles, which are critical for integrating proprioceptive inputs and coordinating motor outputs. Disruption of these pathways, particularly the middle and superior cerebellar peduncles, leads to imbalance and ataxia, hallmark symptoms of cerebellar dysfunction.

In contrast, dysfunction of cranial nerve VIII typically manifests as vestibular symptoms such as vertigo, nystagmus, or hearing loss, none of which were prominently observed in this patient. Additionally, the absence of positional exacerbation of symptoms further excludes a peripheral vestibular cause. This aligns with the well-established understanding that cerebellar lesions impair the processing and modulation of balance rather than the initial sensory input, as would occur with cranial nerve VIII involvement. Given the lesion’s anatomical location and the clinical presentation, it is scientifically accurate to conclude that the imbalance was a consequence of cerebellar peduncle dysfunction, underscoring the necessity of precise localization in correlating clinical symptoms to pathology. The Romberg test showed slight unsteadiness with eyes closed, and tandem gait testing revealed mild instability. Despite these findings, the patient had no nystagmus, and her hearing was intact. Coordination tests, including the finger-to-nose and heel-to-shin tests, were performed smoothly, and there were no signs of dysmetria or limb ataxia. Muscle strength, sensory testing, and reflexes were normal, with no evidence of focal neurological deficits. These findings were consistent with vestibulocerebellar involvement.

Given the patient’s symptoms and neurological findings, an MRI scan of the brain was performed to further evaluate potential cerebellar or vestibular pathology. The MRI revealed a well-circumscribed cavernous malformation (cavernoma) located in the right cerebellar hemisphere (Figure 1 and Figure 2). The lesion appeared hyperintense on T2-weighted imaging, with a peripheral rim of hemosiderin, suggestive of prior microhemorrhages. The lesion measured approximately 1.5 cm in diameter and was localized to the posterior fossa, without any evidence of acute hemorrhage or mass effect. The surrounding cerebellar structures, including the brainstem, showed no significant compression or deviation. Magnetic resonance angiography (MRA), performed to rule out vascular anomalies, confirmed the absence of arteriovenous malformations or other vascular abnormalities (Figure 1D and Figure 2D). The lesion was determined to be isolated, with no direct involvement of the major cerebellar arteries or veins.

A right paramedian craniectomy of the posterior cranial fossa was performed to address a cavernous malformation deeply embedded in the inferior right cerebellar hemisphere. This region, characterized by its dense neurovascular architecture and proximity to critical structures such as the brainstem and cranial nerves, required meticulous surgical planning and execution. The approach was informed by recent advancements in posterior cranial fossa surgery, which prioritize precise exposure, reduced manipulation of surrounding tissue, and preservation of critical functions.

The patient was positioned prone with the head stabilized in a neutral position to optimize surgical access while minimizing venous congestion. Following a thorough review of the lesion’s neuroimaging, a paramedian suboccipital craniectomy was selected for its ability to provide targeted exposure to the lesion while maintaining minimal disruption to surrounding cerebellar structures. This approach reflects contemporary recommendations favoring tailored craniotomies that balance access with safety.

The dura, under tension due to prior microhemorrhages, was incised in a stellate pattern to ensure a controlled opening without compromising adjacent structures. High-magnification microscopy guided the dissection, and neuronavigation was employed to map the lesion’s precise location, minimizing unnecessary manipulation of the cerebellar parenchyma.

The cavernoma, measuring approximately 2.5 cm, was notable for its history of multiple microhemorrhages, as evidenced by hemosiderin deposition. These characteristics, along with their location within the compact cerebellar tissue, demanded careful handling to avoid intraoperative bleeding or neural injury. Drawing from recent literature on posterior fossa cavernomas, a microsurgical technique emphasizing sharp dissection and minimal retraction was used to delineate and resect the lesion en bloc. This method aligns with contemporary strategies aimed at achieving maximal lesion removal with minimal disruption to surrounding anatomy.

Hemostasis was secured using a combination of bipolar cautery and hemostatic agents, in accordance with best practices for vascular lesions in the posterior cranial fossa. A thorough inspection of the surgical cavity confirmed complete resection, an essential factor given that residual cavernoma tissue is strongly associated with a heightened risk of recurrence and rebleeding.

The dura was closed primarily to ensure a watertight seal, preventing cerebrospinal fluid leakage, and an epidural drain was placed to mitigate postoperative complications such as fluid accumulation or hematoma. The surgical approach reflects key principles highlighted in recent literature, including the importance of precise lesion localization, minimal tissue manipulation, and advanced hemostatic techniques in the posterior fossa.

This case illustrates the successful application of modern surgical approaches to manage a challenging posterior fossa lesion. The integration of contemporary techniques, such as tailored craniectomies and advanced visualization tools, underscores the evolution of posterior cranial fossa surgery in reducing patient morbidity while optimizing outcomes.

Postoperatively, the patient experienced a smooth recovery, with no new neurological deficits and significant improvement in preoperative symptoms, particularly vertigo. This outcome highlights the importance of leveraging up-to-date surgical principles and practices in addressing deep-seated lesions in this anatomically complex region.

Neurological assessments were performed regularly, and, notably, no new neurological deficits were detected. The patient remained fully alert and oriented, with intact cranial nerve function, normal coordination, and no signs of cerebellar dysfunction. The vertiginous symptoms that had brought her to the clinic were greatly reduced, with the patient reporting significant improvement in balance and stability by the second postoperative day.

On postoperative day 5, a follow-up CT scan (Figure 3) was performed to assess the surgical site and confirm the absence of postoperative complications such as hemorrhage or hydrocephalus.

The two-month postoperative CT scan (Figure 4) confirmed excellent surgical outcomes. The right cerebellar hemisphere, where the cavernoma had been resected, demonstrated normal postoperative changes, with no signs of residual cavernous malformation or new lesions. The surgical cavity appeared stable, and no signs of hemorrhage or fluid collection were noted.

Clinically, the patient continued to demonstrate full recovery, with no neurological deficits. Coordination tests remained normal, and there was no recurrence of her vertigo or gait instability. Given these findings, the patient was discharged from regular follow-up, with instructions for routine imaging in the future as part of long-term surveillance. At the 1-year follow-up, a control CT scan (Figure 5) was performed to assess the long-term stability of the surgical site and ensure that no delayed complications had arisen.

The patient remained asymptomatic and fully functional at the 1-year mark, with no recurrence of vertiginous symptoms or neurological deficits. Based on these findings, the patient was advised to continue with routine imaging as part of long-term surveillance, but no further immediate follow-up was deemed necessary given the stable postoperative course.

## 3. Discussions

CCMs are rare vascular anomalies that present unique diagnostic and therapeutic challenges, particularly when located in the posterior fossa. The posterior fossa contains critical neurovascular structures, including the brainstem and cranial nerves, making lesions in this region especially concerning due to the potential for catastrophic hemorrhages and significant neurological deficits. This case highlights a successful microsurgical resection of a symptomatic cerebellar cavernoma, resulting in complete resolution of symptoms and no postoperative complications, underscoring the importance of timely surgical intervention and advancements in intraoperative technologies that have improved clinical outcomes.

The patient presented with vertigo, a symptom commonly associated with cerebellar cavernomas, likely due to the lesion’s proximity to cerebellar structures involved in balance and coordination. Symptoms in cerebellar CCMs often result from the mass effect of the lesion or prior hemorrhages, which can lead to secondary effects such as brainstem compression or cranial nerve involvement [7]. Given the patient’s symptomatic presentation and the evidence of prior microhemorrhages, surgical resection was indicated to prevent further neurological deterioration. The high risk of rebleeding in posterior fossa cavernomas, particularly those with a history of hemorrhage, necessitated prompt intervention. According to Gross et al. (2017) [1], patients with prior hemorrhages from CCMs are at increased risk for future bleeds, with annual hemorrhage rates reaching up to 9.8% in posterior fossa lesions. Early surgical intervention has been shown to reduce the risk of subsequent hemorrhage and improve long-term outcomes, making surgery the preferred treatment for symptomatic and hemorrhagic cavernomas [12].

Surgical resection of posterior fossa cavernomas is inherently challenging due to the dense anatomical configuration of the region, which houses critical structures such as cranial nerves and the brainstem [14]. In this case, microsurgical techniques allowed for precise localization and complete resection of the lesion while minimizing manipulation of surrounding tissues. Achieving complete resection, as in this case, is critical to prevent recurrence and rebleeding. Literature [15] emphasized the importance of complete resection, as residual lesion tissue is associated with an increased risk of hemorrhage and persistent symptoms. Although surgery for cerebellar cavernomas carries the risk of neurological deficits due to the lesion’s proximity to essential brain structures, advances in microsurgical techniques and the use of intraoperative monitoring have significantly mitigated these risks. Common postoperative complications include cranial nerve palsies, motor deficits, and brainstem injuries. However, no such deficits were observed in this patient.

Cranial nerve deficits, particularly involving cranial nerves VII (facial) and VIII (vestibulocochlear), are common risks in posterior fossa surgeries. Damage to these nerves can result in facial weakness, hearing loss, or balance dysfunction [16]. Brainstem injuries are a severe potential complication in posterior fossa cavernoma resections, particularly when the lesion is in close proximity to the pons or medulla. Even minor trauma to the brainstem can lead to profound deficits, including motor dysfunction and respiratory compromise. Obstructive hydrocephalus can develop following posterior fossa cavernoma resection due to mass effect on the fourth ventricle or cerebellar swelling [17]. Hydrocephalus may necessitate the placement of a ventriculoperitoneal shunt to relieve increased intracranial pressure. In this case, the absence of postoperative hydrocephalus underscores the importance of careful intraoperative handling of the ventricular system and effective management of postoperative cerebellar edema. Although hydrocephalus remains a concern in posterior fossa surgeries, advancements in surgical techniques and intraoperative management have reduced its incidence.

Long-term follow-up is essential for patients with cavernous malformations, particularly those located in the posterior fossa, due to the risk of recurrence or de novo lesion formation. While complete resection is generally considered curative, regular imaging is recommended to monitor for recurrence or new lesions, especially in patients with familial cavernomatosis. Studies reported that, while patients with complete resection of symptomatic CCMs have a low recurrence rate, up to 10–15% of patients with familial forms of the disease may develop new cavernomas, necessitating regular MRI surveillance [18]. In this case, follow-up CT imaging at two months and one year postoperatively demonstrated a stable resection cavity with no evidence of recurrence or new lesion formation. The patient’s absence of symptoms supports the efficacy of early surgical intervention. However, the two-month follow-up is a recognized limitation, as it does not allow for a full assessment of long-term outcomes, including potential recurrence, de novo lesion formation, or late-onset complications. This shorter duration reflects logistical challenges common in single-case reports, particularly when patients exhibit stable postoperative outcomes without immediate neurological deficits. While early imaging and clinical evaluations confirm the intervention’s success, longer-term follow-up is essential for a comprehensive understanding of surgical outcomes and the natural history of cavernous malformations. Future studies should emphasize extended surveillance, in line with current guidelines recommending annual imaging for the first few years post-surgery, followed by less frequent monitoring of asymptomatic patients.

Although surgical resection remains the gold standard for treating symptomatic or hemorrhagic CCMs, emerging pharmacological treatments show promise for managing CCMs, particularly in non-surgical candidates or patients with multiple lesions. Recent advances in understanding the molecular mechanisms of CCMs have led to the investigation of pharmacotherapies aimed at stabilizing cavernomas and reducing hemorrhage risk [19]. Statins, such as simvastatin, have demonstrated potential in preclinical studies by reducing vascular permeability and endothelial dysfunction. Tang et al. (2017) [20] showed that simvastatin reduced lesion size and vascular leakage in mouse models of CCMs by modulating the Rho kinase signaling pathway, suggesting its utility in CCM management. Propranolol, a beta-blocker commonly used for cardiovascular conditions, has shown potential in reducing cavernoma size and hemorrhage risk. Its proposed mechanism involves the inhibition of angiogenesis and modulation of vascular endothelial growth factor (VEGF), which are implicated in the pathogenesis of cerebral CCMs. However, current evidence remains preliminary, and further studies are needed to establish its efficacy, particularly in symptomatic familial CCM cases [21]. Fasudil, a Rho kinase inhibitor, has also emerged as a potential therapeutic agent. McDonald et al. (2016) [22] demonstrated that fasudil reduces lesion burden and improves vascular stability in genetically modified mouse models of CCMs. While still in the experimental stages, these pharmacological agents may offer new therapeutic options, particularly for patients with familial forms of the disease or inoperable lesions.

Recent advancements in minimally invasive techniques have introduced alternatives to traditional posterior cranial fossa (PCF) approaches, aiming to reduce morbidity while maintaining surgical efficacy [23]. Methods such as the keyhole retrosigmoid approach, endoscopic-assisted microsurgery, and tubular retractor systems offer targeted access with minimal disruption to surrounding tissues [24].

The keyhole retrosigmoid approach, involving a smaller craniotomy, facilitates access to lesions near the cerebellopontine angle and brainstem while reducing operative exposure. Similarly, endoscopic-assisted techniques provide enhanced visualization in confined spaces, minimizing cerebellar retraction [25]. Tubular retractors create a focused surgical corridor, reducing collateral damage to healthy structures. These techniques are particularly effective for smaller, superficial lesions [26].

However, for larger, deeply located, or hemorrhagic lesions, such as the cerebellar cavernoma described here, these minimally invasive methods may not provide the exposure necessary to achieve complete resection and robust hemostasis [27]. In this case, a traditional paramedian craniectomy was chosen to address the lesion’s complexity, offering comprehensive access and ensuring precise control over the surgical field.

While minimally invasive techniques continue to evolve and may expand their applicability in the future, their use must be tailored to lesion size, location, and complexity. For challenging cases requiring extensive dissection and broad visualization, traditional approaches remain indispensable for achieving safe and definitive outcomes.

## 4. Conclusions

This case of a symptomatic cerebellar cavernoma demonstrates the efficacy of early microsurgical resection in preventing further hemorrhage and neurological deterioration. The patient’s favorable outcome, combined with appropriate long-term follow-up, underscores the importance of timely surgical intervention and the role of modern surgical techniques in improving outcomes for patients with posterior fossa cavernomas. Emerging pharmacological treatments, while still under investigation, may provide additional therapeutic options in the future, particularly for patients with familial forms of the disease or inoperable lesions.

## Figures and Tables

**Figure 1 jcm-13-07525-f001:**
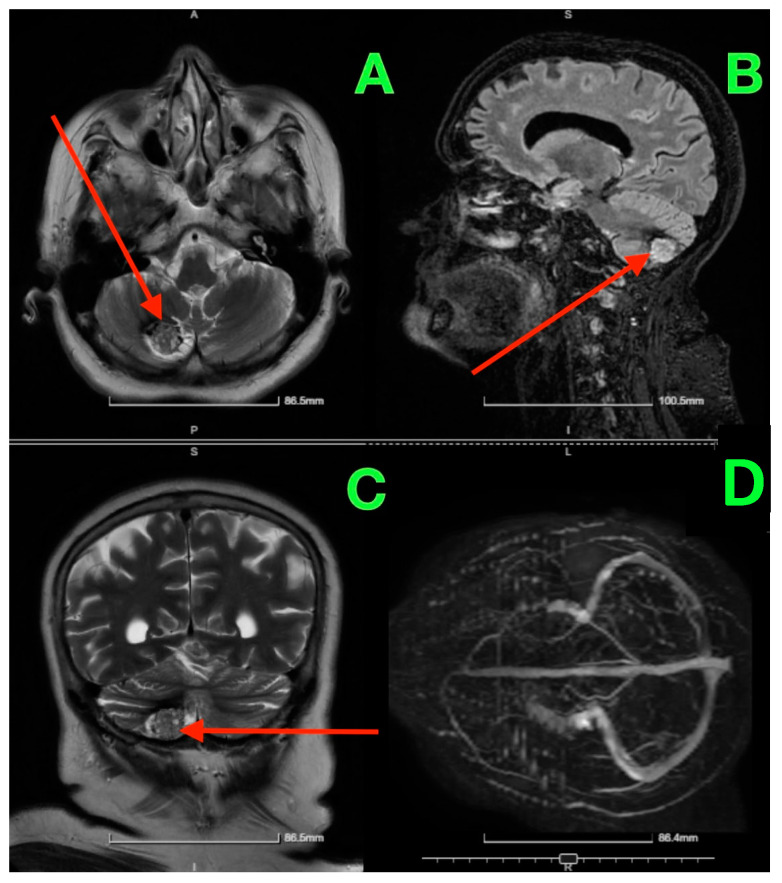
Pre-op MRI scan. (**A**) (Axial T2-weighted MRI): Displays the lesion in the right cerebellar hemisphere, highlighted by the red arrow, which points to a hyperintense core surrounded by hemosiderin deposition at the periphery, consistent with a cavernous malformation. (**B**) (Sagittal T2-weighted MRI): The red arrow indicates the location of the lesion in the posterior fossa, situated near the cerebellar hemisphere, without significant impact on the cerebellar vermis or brainstem. (**C**) (Coronal T2-weighted MRI): The red arrow emphasizes the lesion’s size and well-defined borders relative to the surrounding cerebellar tissue, confirming the absence of surrounding edema. (**D**) (MRA Axial View): No red arrow is present as this image confirms the absence of any vascular abnormalities in proximity to the lesion.

**Figure 2 jcm-13-07525-f002:**
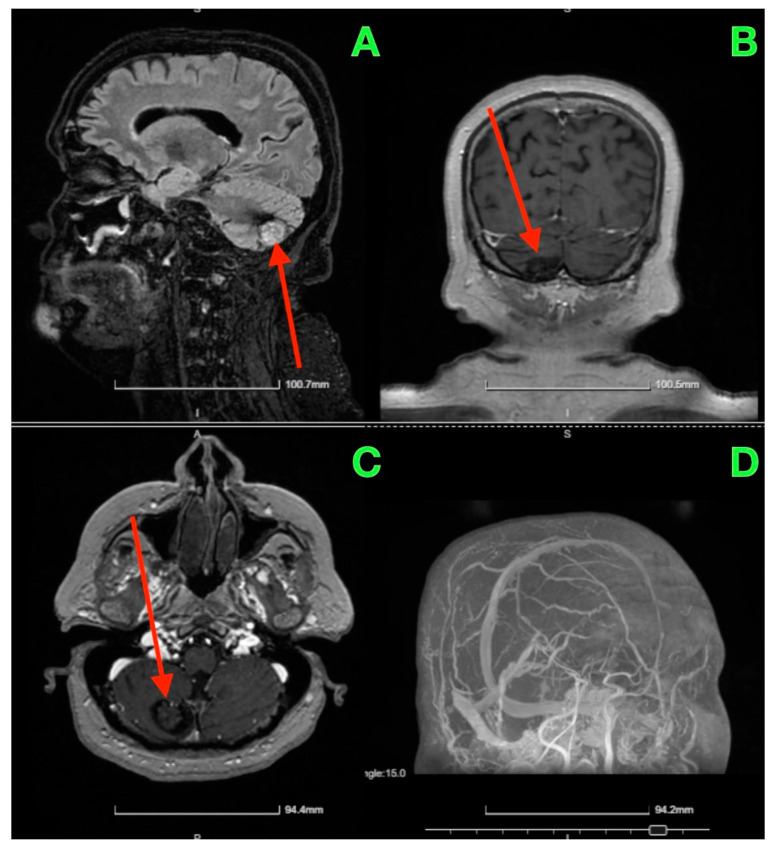
The red arrow highlights the cerebellar cavernous malformation (CCM) located in the right cerebellar hemisphere. This lesion is characterized by its hypointense signal on T1-weighted MRI images (**A**–**C**), consistent with prior microhemorrhages and hemosiderin deposition surrounding the malformation. The lesion’s distinct margins and location, without significant mass effect on adjacent structures such as the brainstem and fourth ventricle, are emphasized by the arrow. The MRA (**D**) further confirms the absence of vascular anomalies, distinguishing the lesion as a solitary cavernoma without associated arteriovenous malformation.

**Figure 3 jcm-13-07525-f003:**
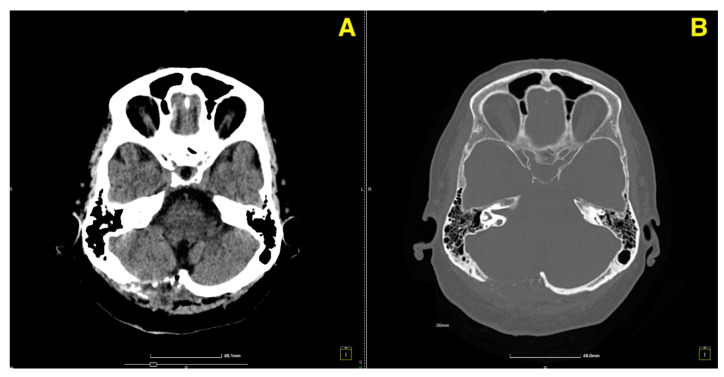
Five days post-op CT scan. (**A**) Demonstrated normal postoperative changes in the right paramedian posterior fossa. The site of the craniectomy was clearly visible, and no signs of residual cavernous malformation were observed. There was no evidence of acute hemorrhage, mass effect, or significant edema in the surrounding cerebellar tissue. Additionally, the ventricular system appeared normal, with no signs of hydrocephalus or midline shift. (**B**) provided further confirmation of the integrity of the bony structures and the absence of any complications related to the craniectomy. The surgical site remained stable, and there was no abnormal fluid collection or air entrapment within the resection cavity.

**Figure 4 jcm-13-07525-f004:**
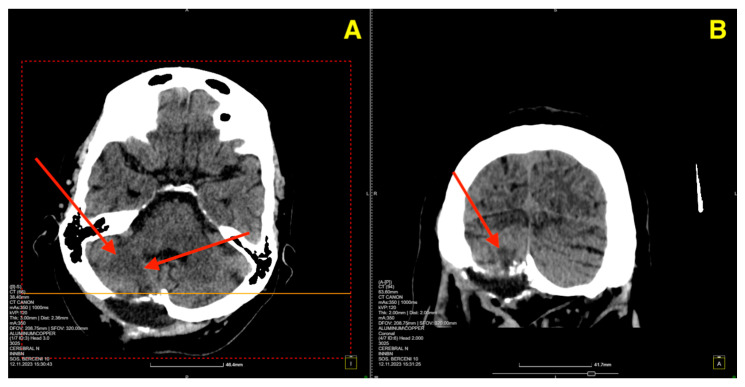
Two-month follow-up post-op CT scan. (**A**): Axial CT image of the right cerebellar hemisphere shows normal postoperative changes (red arrows) in the area of the cavernoma resection, with no residual cavernous malformation, new lesions, or signs of fluid collection or hemorrhage. (**B**): Coronal CT image further confirms the stability of the surgical cavity (red arrow), with no evidence of hemorrhage, recurrent lesions, or abnormal findings in the surrounding structures.

**Figure 5 jcm-13-07525-f005:**
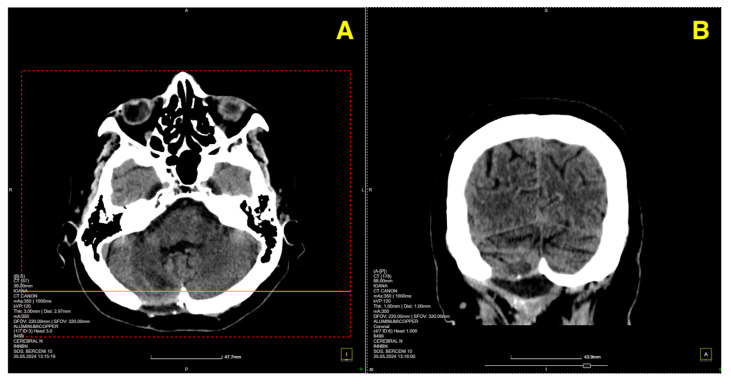
One-year follow-up control CT. (**A**): Axial CT image displays the postoperative region in the right cerebellar hemisphere, showing well-preserved stability with no residual cavernoma or evidence of delayed complications, such as hemorrhage or fluid accumulation. The surrounding cerebellar tissue remains intact and unaffected. (**B**): Coronal CT image highlights the absence of recurrent lesions or structural abnormalities at the surgical site. The cavity remains unchanged, reflecting long-term postoperative success without any signs of new pathological findings.

## Data Availability

The data presented in this study are available on request from the corresponding author.

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
