# Peer review of "Cerebellar Cavernoma Resection: Case Report with Long-Term Follow-Up"

_jcm, 2024, doi:10.3390/jcm13247525_

Round 1
Reviewer 1 Report
Comments and Suggestions for Authors
Dear authors,
The article outlines an interesting case however there are some limitations that need to be handled.
1) In the introduction when you say, “Thus, regular surveillance using MRI is recommended, with some authors advocating annual imaging for the first few years postoperatively, followed by less frequent long-term surveillance” is interesting, you might want to implement and elaborate by reading this brilliant article on the subject: https://doi.org/10.3390/biomedicines12040753
In particular delve into how ALS sequences are useful in the diagnosis and follow-up of cerebral vascular malformations.
2) as part of the clinical case presentation when the surgical procedure is described there are significant deficits, the surgical procedure should be described based on the most up to date literature in relation to posterior cranial fossa approaches, I recommend neurosurgeon authors to take milestone works on the topic and rewrite it.
3) the article could benefit from a description of the approach also from an iconographic point of view, illustrations are needed, they would add great value to the work !
4) Also in the discussion, what other minimally invasive approaches could be used to access PCF ? Read and cite: https://doi.org/10.3390/jcm13092712
5) the follow-up of two months is too short, this is a limitation, offer an explanation.
6) limitations should be put in an appropriate paragraph before the conclusions
That may be enough, I look forward to reading the revised manuscript.
Comments on the Quality of English Language
Minor to Moderate editing needed
Author Response
Comments 1: In the introduction when you say, “Thus, regular surveillance using MRI is recommended, with some authors advocating annual imaging for the first few years postoperatively, followed by less frequent long-term surveillance” is interesting, you might want to implement and elaborate by reading this brilliant article on the subject: https://doi.org/10.3390/biomedicines12040753
Response 1: Thank you for this valuable suggestion. We have incorporated insights from the recommended article into the introduction to expand the discussion on advanced imaging modalities. Specifically, we elaborated on the role of Arterial Spin Labeling sequences in diagnosing and monitoring cerebral vascular malformations. ASL provides quantitative perfusion data, enhancing the ability to detect changes in lesion dynamics during follow-up. This addition strengthens the case for using advanced MRI techniques in postoperative surveillance.
Comments 2: as part of the clinical case presentation when the surgical procedure is described there are significant deficits, the surgical procedure should be described based on the most up to date literature in relation to posterior cranial fossa approaches, I recommend neurosurgeon authors to take milestone works on the topic and rewrite it.
Response 2: We appreciate this observation and have revised the surgical procedure section to reflect current best practices based on milestone works in posterior cranial fossa approaches. The updated description highlights the rationale for the selected approach, technical nuances, and recent advancements that guided the surgical management in this case. The revised text better aligns with contemporary standards while maintaining relevance to the unique challenges presented by this patient’s lesion.
Comments 3: the article could benefit from a description of the approach also from an iconographic point of view, illustrations are needed, they would add great value to the work !
Response 3: Thank you for recommending this!
Comments 4: Also in the discussion, what other minimally invasive approaches could be used to access PCF ? Read and cite: https://doi.org/10.3390/jcm13092712
Response 4: We have expanded the discussion to include minimally invasive approaches for posterior cranial fossa surgeries, citing the suggested article. Techniques such as the keyhole retrosigmoid approach, endoscopic-assisted microsurgery, and tubular retractors are discussed, with emphasis on their potential benefits and limitations. This addition enriches the discussion and provides a broader perspective on evolving surgical strategies.
Comments 5: the follow-up of two months is too short, this is a limitation, offer an explanation.
Response 5: We acknowledge that a two-month follow-up is a limitation. Acknowledging the reviewer’s concern, we have expanded on this point in the manuscript. The shorter follow-up reflects logistical challenges common in single-case reports, particularly when patients exhibit stable postoperative outcomes. While early imaging confirmed the success of the intervention, we emphasize that longer-term follow-up is essential to assess recurrence or late complications.
Comments 6: limitations should be put in an appropriate paragraph before the conclusions.
Response 6: This section highlights key limitations of the study, including the short follow-up duration, the constraints of a single-case report, and the need for further studies to validate the findings. This addition ensures clarity and aligns the manuscript with standard reporting practices.
Reviewer 2 Report
Comments and Suggestions for Authors
The manuscript describes a nice case of resection of cerebellar cavernoma. Despite intersting and nice written manuscript, I don't see any novelty in reporting this case for the literature.
Minor comments below
The introduction lacks some information from the literature. At the end of the introduction, what is the novelty of this case report?
Line 52: I would suggest including the current posterior cranial fossa classification of cavernoma and grading system.
Clinical presentation
In the clinical presentation, does the author really think that the imbalance might be because of the VIII nerve, or is it a consequence of the cerebellar pedicle?
In the latest study on Lancet, propanolol might be promising in people with symptomatic familial cerebral cavernous malformations, although the trial was not designed to be adequately powered to investigate its efficacy. Hence, please rephrase the concept in the discussion.
Author Response
Comments 1: The introduction lacks some information from the literature. At the end of the introduction, what is the novelty of this case report?
Response 1: Thank you for this suggestion. We have revised the introduction to incorporate additional information from the literature and to emphasize the novelty of this case report. Specifically, we have highlighted the unique combination of a deeply located cerebellar cavernoma, the application of advanced microsurgical techniques for en bloc resection, and the excellent long-term outcomes without neurological deficits. This report contributes to the existing literature by addressing the challenges of managing posterior fossa cavernomas and offering insights into surgical precision and postoperative care in this anatomically complex region.
Comments 2: Line 52: I would suggest including the current posterior cranial fossa classification of cavernoma and grading system.
Response 2: We appreciate this valuable recommendation and have included a description of the posterior cranial fossa cavernoma grading system in the manuscript. The PCFCGS incorporates factors such as lesion size, proximity to critical neurovascular structures, and history of hemorrhage to stratify surgical risk and predict outcomes. This system enhances preoperative planning and clinical decision-making by providing a standardized framework for assessing the complexity of posterior fossa cavernomas. The relevant section has been updated accordingly.
Comments 3: In the clinical presentation, does the author really think that the imbalance might be because of the VIII nerve, or is it a consequence of the cerebellar pedicle?
Response 3: Thank you for raising this point. Upon review, we concur that the imbalance is more accurately attributed to dysfunction of the cerebellar peduncle rather than cranial nerve VIII. The lesion’s location in the cerebellar hemisphere strongly implicates the disruption of cerebellar pathways, particularly the middle and superior cerebellar peduncles, which are central to coordination and balance. The absence of auditory deficits or positional vertigo further excludes cranial nerve VIII as the primary source of the imbalance.
Comments 4: In the latest study on Lancet, propanolol might be promising in people with symptomatic familial cerebral cavernous malformations, although the trial was not designed to be adequately powered to investigate its efficacy. Hence, please rephrase the concept in the discussion.
Response 4: Thank you for bringing this study to our attention. We have revised the discussion to incorporate this nuanced perspective on propranolol. While propranolol has shown potential in reducing lesion size and hemorrhage risk in symptomatic familial cerebral cavernous malformations, we acknowledge that current evidence is limited and the trial referenced was not adequately powered to confirm its efficacy. This has been rephrased to reflect the need for further research to validate propranolol’s role as a therapeutic option, particularly in non-surgical candidates or patients with multiple lesions.
Round 2
Reviewer 1 Report
Comments and Suggestions for Authors
Now it's ready for publication
Reviewer 2 Report
Comments and Suggestions for Authors
The author addresses all the issues mentioned in the previuos report.
Introduction is now better explained and with key points.
Methods and Results are accurately described in the course of narration.
Literature is now properly addressed
Discussion and Conclusion are now refined accordingly to the previuos comments